# CD79A and IL7R mRNA Levels in the Cerebrospinal Fluid of Adults with Acute B-Cell Lymphoblastic Leukemia: A Pilot Study

**DOI:** 10.3390/diseases13070206

**Published:** 2025-07-01

**Authors:** Andrea Iracema Milán Salvatierra, Juan Carlos Bravata Alcántara, Víctor Manuel Alvarado Castro, Estibeyesbo Said Plascencia Nieto, Faustino Cruz Leyto, Mónica Tejeda Romero, Jorge Cruz Rico, Bogar Pineda Terreros, Sandra López Palafox, Adriana Jiménez, Juan Ramón Padilla Mendoza, José Bonilla Delgado, Catalina Flores-Maldonado, Enoc Mariano Cortés Malagón

**Affiliations:** 1Hematology Service, Hospital Juárez de México, Mexico City 07760, Mexico; doc.milandrea@gmail.com (A.I.M.S.); fleyto@yahoo.com.mx (F.C.L.); trmonica@gmail.com (M.T.R.); jcruzrico@gmail.com (J.C.R.); bogarpineda@gmail.com (B.P.T.); sandralopzpal@gmail.com (S.L.P.); 2Genetics and Molecular Diagnostics Laboratory, Hospital Juárez de México, Mexico City 07760, Mexico; vaio_df@hotmail.com; 3Centro de Investigación de Enfermedades Tropicales, Universidad Autónoma de Guerrero, Acapulco 39630, Mexico; alvarado@cimat.mx; 4Sección de Estudios de Posgrado e Investigación, Escuela Superior de Medicina, Instituto Politécnico Nacional, Mexico City 11340, Mexico; saidpn@yahoo.com.mx; 5Research Division, Hospital Juárez de México, Mexico City 07760, Mexico; adijh@hotmail.com; 6Laboratory of Cellular Reprogramming, Department of Physiology, Facultad de Medicina, Universidad Nacional Autónoma de México, Mexico City 04510, Mexico; juan.padilla@cinvestav.mx; 7Laboratory of Molecular Oncology, Hospital Regional de Alta Especialidad de Ixtapaluca, Servicios de Salud del Instituto Mexicano del Seguro Social para el Bienestar (IMSS-BINESTAR), Ixtapaluca 56530, Mexico; jose.bonillad@hraei.gob; 8Department of Physiology, Biophysics and Neurosciences, Center for Research and Advanced Studies of the Instituto Politécnico Nacional (Cinvestav-IPN), Mexico City 07360, Mexico; catalina.flores@cinvestav.mx; 9Genetics Laboratory, Hospital Nacional Homeopático, Mexico City 06800, Mexico

**Keywords:** CD79A, IL7R, mRNA, acute B-cell lymphoblastic leukemia, CNS

## Abstract

Background/Objectives: In adults with B-cell acute lymphoblastic leukemia (B-ALL), central nervous system (CNS) involvement represents a significant clinical challenge due to its association with adverse outcomes. Infiltration of blast cells into the CNS is primarily detected via cerebrospinal fluid (CSF) microscopy, the current gold standard diagnostic method, although it has limitations in terms of sensitivity. Quantitative polymerase chain reaction (qPCR) offers higher sensitivity and can support the diagnosis of CNS infiltration. This study assessed the mRNA expression levels of CD79A and IL7R in CSF to evaluate their potential for detecting CNS involvement in adults with B-ALL. Methods: CSF samples were collected from adults with B-ALL. The classification criteria for CNS Leukemia (CNS status) were used to evaluate CNS involvement. RNA was extracted from the CSF, and quantitative reverse transcription PCR (RT-qPCR) was used to measure the CD79A and IL7R mRNA expression levels. Results: A total of 19 treatment-naïve adult patients with B-ALL were enrolled over a 19-month period. Four (21%) patients had CNS3 status. Four (21%) patients had CNS3 status. The results also showed that the expression levels of CD79A and IL7R mRNA were significantly higher (median fold change = 0.62 and 2.12, *p* < 0.05, respectively) in the group with CNS3. Furthermore, using the Haldane-Anscombe correction and Fisher’s exact test, we demonstrated an association between IL7R and CNS3 expression (odds ratio = ∞, due to zero CNS+ in the IL7R group, *p* < 0.05). Conclusions: CD79A and IL7R mRNA levels in CSF could be potential biomarkers for detecting CNS involvement in adult patients with B-ALL.

## 1. Introduction

B-cell acute lymphoblastic leukemia (B-ALL) is a malignant hematologic disorder characterized by the rapid proliferation of immature B-cell precursors in the bone marrow, peripheral blood, and extramedullary sites [1]. Despite advances in treatment strategies, the prognosis for adult B-ALL patients remains less favorable than that for pediatric patients, with a significantly lower 5-year progression-free survival rate in adults [2]. Unlike children, adults more commonly exhibit the Philadelphia chromosome or abnormalities associated with poor prognosis [3,4].

The involvement of the CNS in B-ALL is a major clinical concern because it is associated with poor prognosis and a higher risk of relapse. Once leukemic cells enter the CNS, they can evade the effects of systemic chemotherapy due to the protective function of the blood-brain barrier [5]. Identifying CNS disease at an early stage allows for the timely initiation or intensification of CNS-directed therapies, including intrathecal chemotherapy or high-dose systemic regimens. This approach has been shown to improve disease control, reduce the probability of CNS relapse, and enhance overall survival [6].

Leukemic infiltration into the CNS is thought to occur through multiple, yet incompletely understood mechanisms. Proposed routes include trans-endothelial migration across the blood-brain barrier, entry via the choroid plexus, or trafficking along leptomeningeal or perivascular spaces. However, direct experimental evidence supporting these pathways remains limited and largely model-dependent [7,8]

In B-ALL, specific cytogenetic abnormalities are strongly linked to a higher risk of CNS leukemia (CNSL). Among these, the t(1;19) translocation, which gives rise to the E2A-PBX1 fusion, and the t(9;22) translocation, responsible for the BCR-ABL fusion (also known as the Philadelphia chromosome), are particularly associated with CNS infiltration [9,10]. Notably, the E2A-PBX1 fusion has also been linked to the upregulation of the interleukin-7 receptor (IL7R), suggesting a potential molecular connection between leukemic signaling pathways and CNS tropism [10]. Adding to this complexity, recent findings have shown elevated expression of the IL7Rα chain (CD127)—a key subunit of IL7R—in B-ALL patients harboring TP53 mutations and CRLF2 rearrangements [11], further highlighting the potential relevance of IL7R signaling in high-risk disease and CNS involvement.

The IL7R and pre-BCR signaling pathways intersect at several key points, affecting B cell development and leukemia. Both pathways activate similar downstream molecules, such as PI3Kalpha/PKB and RAS/ERK, which are essential for cell growth and survival [12]. B cells develop properly by combining signals from IL7R and pre-BCR; however, when these signals are disrupted in B-ALL, they lead to the sustained growth of leukemia and actively drive leukemogenesis [13,14]. Recent studies have provided compelling evidence that IL7R activation supports leukemic cell expansion and acts as an initiating oncogenic driver, thereby contributing directly to leukemogenesis [15,16,17].

Moreover, the B-cell antigen receptor complex-associated protein alpha chain (CD79A) is a critical component of the pre-BCR signaling pathway, a lineage-specific marker of B lymphoid cells, and plays an important role in the development and diagnosis of ALL [18].

Several methods are available for detecting leukemia infiltration or cell tumor infiltration in the CNS. Cytometric analysis using cytospin or multicolor flow cytometry has been employed to detect leukemic blasts in the CSF [19,20]. However, cytospin is limited by its low sensitivity and reproducibility [21,22]. Although CSF evaluation via flow cytometry (FCM) is better than cytomorphology via cytospin, FCM requires intact cells for analysis. Therefore, the sensitivity of this analysis decreases with the degradation of lymphocytes in the CSF [20]. In addition, FCM requires specialized equipment and expertise, and the interpretation of the results can be complex, making FCM challenging to use in low- and middle-income countries [23].

Advancements in molecular biology have led to improved methods for detecting RNA in the central nervous system (CNS). Researchers have analyzed the RNA composition of extracellular vesicles and the total RNA associated with various neurological disorders, such as Alzheimer’s disease, Parkinson’s disease, low-grade glioma, glioblastoma multiforme, and subarachnoid hemorrhage [24]. Furthermore, quantifying mRNA in the CSF has become valuable for assessing CNS metastases. Notably, this method identified elevated levels of CEACAM6 mRNA in the CSF of patients with lung cancer-associated leptomeningeal metastases [25]. Previous studies have also demonstrated the potential of PCR-based techniques for detecting leukemic involvement in the CSF of patients with ALL, particularly in the pediatric population. Notably, Pine et al. demonstrated the feasibility of using qPCR to detect leukemic blasts in the CSF of pediatric patients with ALL. They found that PCR could identify leukemic cells even in cases where traditional cytological methods failed, highlighting the sensitivity of qPCR in detecting CNS involvement [26]. Similarly, Péterffy et al. used digital PCR to quantify microRNA-181a in CSF samples, enabling the stratification of pediatric ALL patients according to CNS involvement and highlighting the diagnostic potential of RNA-based biomarkers [27]. Furthermore, other studies have utilized patient-specific primers targeting immunoglobulin and T-cell receptor gene rearrangements in CSF DNA, demonstrating that PCR can detect submicroscopic levels of leukemic cell infiltration with higher sensitivity than that of cytospin techniques [28]. Collectively, these findings support the utility of PCR-based strategies as complementary tools for the early detection and monitoring of CNS diseases in ALL.

Research on the presence of IL7R and CD79A mRNAs in the CSF in cases of metastasis is limited. However, studies have suggested a relationship between IL7R or CD79A expression and CNS infiltration in certain oncological conditions. A study investigated IL7R expression in pediatric B-ALL patients. The results revealed that higher IL7R levels in bone marrow leukemic cells were associated with CNS infiltration at diagnosis and a higher risk of CNS relapse. A xenograft model in immunodeficient mice showed that IL7R is crucial for CNS infiltration, and that using a monoclonal antibody against IL7R can reduce this infiltration [29]. Another study indicated that CD79A might contribute to CNS infiltration in pediatric B-ALL, as dysregulated CD79A signaling can enhance the migratory and invasive properties of leukemic cells, facilitating their entry into the CNS [30].

Although these studies did not directly evaluate the presence of IL7R or CD79A mRNAs in the CSF, their findings indicate an association between their expression and CNS infiltration in pediatric B-ALL. Further investigations are required to determine whether IL7R and CD79A mRNAs are present in the CSF of patients with metastasis and whether they can serve as biomarkers.

Therefore, our study aimed to investigate the presence and mRNA levels of CD79A and IL7R in the CSF of adults with B-ALL using qPCR, which may provide a more sensitive and precise method for the early detection of CNS metastases than traditional methods.

## 2. Materials and Methods

### 2.1. Study Design and Patients

This research was a prospective cross-sectional observational pilot study. This study followed the Strengthening the Reporting of Observational Studies in Epidemiology (STROBE) guidelines and received approval from the ethical and research committees (HJM 024/21-1) at Hospital Juárez de México on 18 October 2021. This study enrolled 19 treatment-naïve B-ALL patients (≥18 years old) between December 2021 and June 2023. The inclusion criteria were as follows: newly diagnosed B-ALL according to the WHO classification criteria; availability of CSF sample collected at diagnosis or prior to the initiation of systemic or intrathecal chemotherapy; and complete clinical records, including neurological assessment, CSF cytology, and MRI. Patients excluded from the study were those with a history of CNS pathology unrelated to leukemia (e.g., CNS infections, neurological disorders, or brain tumors), a diagnosis of mixed phenotype acute leukemia or non-B-lineage ALL, or insufficient RNA yield or quality for qPCR analysis.

The sample size was based on the availability of patients. All participants provided written informed consent after receiving comprehensive written and oral explanations of the study. Two independent investigators reviewed and verified the data to ensure consistency and minimize the extraction errors. Discrepancies were resolved through consensus.

### 2.2. B-ALL Diagnosis

Diagnostic tests for identifying type B-ALL were carried out by experienced professionals at the Clinical Analysis Laboratory at Hospital Juárez de México. The percentage of bone marrow blasts needed to be ≥20% and was classified according to the French American British (FAB) system [31]. In addition, EuroFlow immunophenotyping [32] was used to classify B-ALL using the following antibody panel: CD3s, CD10, CD19, CD20, CD22, CD34, CD38, CD45, CD58, CD66c, CD79a, CD123, TdT, IgM, Kappa, and lambda. Furthermore, the LEUKEMIA Fusion Genes (Q30) Screening Kit (QuanDx, San Jose, CA, USA) was used to detect 30 characteristic fusion genes of acute leukemia. Standard clinical laboratory tests were also performed.

### 2.3. Cerebrospinal Fluid Collection

The procedure involved a lumbar puncture (LP) performed by qualified physicians to collect the CSF. Initially, LP was performed between the third and fourth lumbar vertebrae. Once the space was located, 3 to 5 mL of 1% lidocaine local anesthetic was injected into the subcutaneous tissue and left for 5 min. A 20-gauge LP needle was subsequently inserted in the midline, equidistant from the two processes, and slowly advanced while maintaining a horizontal angle parallel to the dural fibers. Three milliliters of CSF were collected and divided into 1 mL for cytomorphology and 2 mL for molecular techniques. The CSF for cytomorphology was then centrifuged at 400× *g* for 10 min at room temperature and immediately analyzed. The CSF used in the molecular approach was aliquoted into volumes of 1.0 mL and stored at −80 °C until needed [33].

### 2.4. CSF Cytology

For cytomorphology, 1 mL of CSF was centrifuged at 400× *g*. Cell counts were manually evaluated using a Neubauer chamber. In cases where blasts are present and the red blood cell to white blood cell (RBC/WBC) ratio is ≤100:1, the samples should be analyzed via cytocentrifugation. This RBC/WBC ratio suggests that the lumbar puncture was nontraumatic and that the CSF was free from blood contamination [28].

### 2.5. Clinical Signs of CNS Leukemia

Some patients may show neurological symptoms, including headache, nausea, and vomiting (due to increased intracranial pressure). Cranial nerve palsies (e.g., facial weakness and vision changes). Seizures or altered mental status may occur in severe cases. Meningeal signs (stiff neck and photophobia) are rare.

### 2.6. MRI-Based Assessment of CNS Involvement

CNS involvement was evaluated using MRI scans (GE HealthCare Technologies, Inc., Chicago, IL, USA) of the brain and/or spine. The MRI findings that were considered indicative of leukemic infiltration included (1) leptomeningeal enhancement, (2) focal or diffuse parenchymal lesions not attributable to any other cause, and (3) abnormalities in the ventricles or subependymal regions suggesting leukemic spread. All MRI scans were independently reviewed by board-certified radiologists who were unaware of the patients’ molecular and cytological results. Radiologic assessments were interpreted in conjunction with clinical data to support CNS classification, particularly in cases where the cytological results were negative.

### 2.7. Diagnostic Criteria for CNS Leukemia

The diagnostic criteria are established based on cytological, clinical, and MRI findings (Table 1). The most widely accepted classification follows guidelines from organizations such as the National Comprehensive Cancer Network (NCCN), World Health Organization (WHO), and research groups, including the Children’s Oncology Group (COG.) CNS involvement in leukemia is classified as CNS1, CNS2, and CNS3 based on the CSF analysis [28,34].

### 2.8. RNA Extraction

Total RNA was extracted from the CSF samples using the mirVana Paris miRNA Isolation Kit (Invitrogen, Waltham, MA, USA) according to the manufacturer’s instructions. Initially, 625 μL of CSF was combined with an equal volume of 2× denaturing solution at RT. An equal volume of acid‒phenol‒chloroform was added to the sample lysate mixture and 2× denaturing solution. The mixture was centrifuged at 10,000× *g* for 5 min at RT, and the aqueous phase was carefully separated. A total of 1.25 volumes of 100% ethanol was added to the aqueous phase, and the resulting solution was transferred to a filter cartridge and centrifuged at 10,000× *g* for 30 s at RT. The flow-through was discarded, and the process was repeated until the entire lysate/ethanol mixture passed through the filter. Next, 700 μL of miRNA wash solution was added to the filter cartridge and centrifuged at 10,000× *g* for 15 s at RT. Then, 500 μL of 2/3 wash solution was applied to the filter cartridge and centrifuged as described previously. To recover the RNA, 100 μL of preheated (95 °C) elution solution or nuclease-free water was added to the filter, which was subsequently centrifuged at 10,000× *g* for 30 s at RT, collected in a nuclease-free tube, and stored at −80 °C until use.

To maximize RNA recovery through repeated extraction [35], we rehydrated the remaining interphase and organic phase and re-extracted the phenol‒chloroform solution with water, following the above procedure. The concentration and purity of the RNA were assessed using a NanoDrop spectrophotometer (Thermo Fisher Scientific, Waltham, MA, USA). Only RNA samples with a quality index (260 nm/280 nm absorbance ratios between 1.8 and 2.2 and 260 nm/230 nm absorbance ratios greater than 1.8) were included in the study. RNA integrity was assessed using a Qsep1 Bio-Fragment Analyzer (BiOptic Inc., New Taipei City, Taiwan) for RNA, according to the manufacturer’s instructions.

### 2.9. RT‒qPCR

Quantitative PCR (qPCR) was performed to quantify the expression levels of CD79A and IL7R mRNAs. cDNA was synthesized from 100 ng of total RNA using the LunaScript RT SuperMix Kit (New England Biolabs, Ipswich, MA, USA) according to the manufacturer’s instructions. Negative controls, such as reactions without retrotranscriptase or templates, were included. The cDNA synthesis reaction was carried out in a thermal cycler at 25 °C for 2 min, followed by 55 °C for 10 min, and then 95 °C for 5 min to inactivate the reverse transcriptase.

qPCR was performed in a 20 μL volume containing 2× Luna Universal Probe qPCR Master Mix (New England Biolabs, Ipswich, MA, USA), 0.4 μM each of forward and reverse primers, 0.2 μM probe, 100 ng of cDNA template, and RNase-free water. The primer sequences used for CD79A, IL7R, the probe, and the reference gene GAPDH are listed in Table 2. qPCR was performed on a StepOne Real-Time PCR System (Applied Biosystems, Foster City, CA, USA) under the following conditions: initial denaturation at 95 °C for 60 s, followed by 45 cycles of 95 °C for 15 s and 60 °C for 30 s. A threshold cycle (Ct) value of < 40 was considered positive. Melt curve analysis was performed to confirm the specificity of the amplification products. All reactions were performed in triplicate.

The relative expression levels of CD79A and IL7R were calculated using the 2^−ΔCt^ method [36], with GAPDH serving as the reference control.

### 2.10. Statistical Analysis

A univariate analysis was performed to obtain descriptive statistics, such as simple frequencies of the study variables and measures of central tendency. The Gaussian distribution was tested using the Shapiro–Wilk test. To compare the Cts of CD79A, IL7R, and GAPDH, the Kruskal–Wallis test and Dunn’s test were used as post hoc tests. mRNA levels were assessed using the Wilcoxon test. A *p*-value < 0.05 was considered significant. Associations between several factors, including gene expression and CNS status, in patients with B-ALL were examined using bivariate analysis to calculate the odds ratio (OR) and 95% CI.

All the statistical analyses were performed via R (version 4.0.3). Graphs were created using GraphPad Prism (version 9.0, GraphPad Software).

## 3. Results

Between December 2021 and June 2023, 19 treatment-naïve B-ALL patients were enrolled. A total of 47.4% (9/19) were females and 52.6% (10/19) were males, with a mean age of 33 ± 15.4 years (range, 18–67). The average body mass index (BMI) was 27.5 ± 5.1 kg/m^2^, with a range of 18.0–38.5. In terms of clinical laboratory data, varying values were noted, including leukocytosis to leukopenia [55,477 ± 90,431/mm^3^ (1040–298,830)], hemoglobin levels of 6.3 g/dL (3.60–12.7), and platelet counts of 44,386/μL (1460–309,000). These measurements provided a detailed view of the patients’ physical condition at the time of B-ALL diagnosis (see Table 3 for details).

For the analysis of cellular phenotypes, we used Wright’s staining to examine peripheral blood samples. The results revealed that, on average, 55.2 ± 33.5% of lymphoid blasts were present. However, one patient had no blasts detected, whereas the other two had two and three blasts in their peripheral blood, respectively. In contrast, the blast count in the bone marrow was consistent, with an average of 86.3 ± 16.5% lymphoid blasts present. They were classified using the FAB classification system, with L2 comprising 84.2% (16 cases) and L1 comprising 15.8% (3 cases). Additionally, according to the EuroFlow guidelines, FCM was used for leukemia classification. A total of 89.5% of patients had cells classified as pre-B, whereas 5.3% were classified as common-B, and 5.3% were classified as pro-B (Appendix A).

A panel of 30 leukemia fusion genes was analyzed in the bone marrow to assess the molecular risk. Only one patient with the 9; 22 translocation (BCR-ABL) was identified.

Clinical evaluation of CNS infiltration was conducted through patient interviews to identify potential signs of CNS involvement. Remarkably, only a small subset of patients exhibited symptomatic manifestations: four patients reported severe, persistent headaches unresponsive to analgesics, and one patient presented with reduced visual acuity. Lumbar puncture was performed in patients with B-ALL to evaluate CNS blast cell infiltration. During this process, 3 mL of CSF was collected, and cytomorphological examination was conducted to search for blasts. This examination revealed the presence of blasts in three patients at the time of diagnosis, which represented 15.8% of the cases. Among these patients, only two patients (2/19, 11%) met the criterion of having ≥5 blasts/mm^3^.

Magnetic resonance imaging (MRI) has been utilized for its exceptional capacity to visualize the brain and spinal structures in detail, making it an essential tool for assessing the extent of disease infiltration. In our study, MRI identified evidence of CNS infiltration in 21% (4 of 19) of patients, characterized by meningeal enhancement and thickening, hyperintense lesions, and contrast enhancement. These imaging features indicate leukemic infiltration and reflect varying degrees of CNS involvement. Finally, according to the classification of CNS leukemia (Table 3), four patients were classified as CNS3.

We performed RT-qPCR on CSF samples from patients with B-ALL to assess the potential of CD79A and IL7R mRNAs as biomarkers for CNS infiltration. Among the samples analyzed, 52.6% (10/19) were positive for CD79A mRNA, and 42.1% (8/19) were positive for IL7R mRNA. Furthermore, 26.3% (5/19) of samples were positive for both biomarkers (Figure 1A and Appendix A). Importantly, positive Ct values for GAPDH were detected in all CSF samples. Comparative analysis of the mRNA expression levels revealed that IL7R expression was greater than that of CD79A, with a mean fold change of 1.25 ± 1.06 for IL7R and 0.51 ± 0.62 for CD79A (Figure 1B and Appendix A).

The expression levels of CD79A and IL7R in the CSF were analyzed and compared between B-ALL patients with and without CNS involvement, as determined by MRI findings. CNS involvement was defined as the presence of lesions or abnormalities detected on MRI. Patients with MRI-detected CNS involvement presented slightly elevated levels of CD79A expression in the CSF, with a median of 0.62, whereas those without CNS involvement had a median of 0.19 (Figure 2A). Additionally, IL7R expression levels were significantly higher in the CNS-positive group, with a median fold change of 2.12, than in the CNS-negative group, with a median fold change of 0.38 (Figure 2B).

We also conducted a bivariate analysis to evaluate the potential factors associated with CNS infiltration in patients with B-ALL (Table 4). However, none of the clinical factors examined showed a statistically significant association with CNS infiltration. Our analysis revealed an association between IL7R mRNA expression in CSF and CNS involvement in adult B-ALL patients. None of the IL7R-negative cases exhibited CNS infiltration, resulting in an infinite odds ratio after applying the Haldane–Anscombe correction. This association was statistically significant (*p* = 0.018, Fisher’s exact test). In contrast, although CD79A expression also yielded an infinite odds ratio under the same correction, the association did not reach statistical significance (*p* = 0.087).

## 4. Discussion

In B-ALL, involvement of the CNS is a major clinical concern because it is linked to poorer prognosis and a greater risk of relapse. Assessment of blast cell CNS infiltration is typically performed using CSF microscopy, which remains the preferred method despite its limitations in sensitivity and the amount of information it provides [37]. Alternative techniques are a reliable complement to cytomorphology, and their combination can significantly enhance the detection of blast cell CNS involvement [21,37].

The examination of CSF is the most valuable test for diagnosing CNS involvement. Abnormal findings may include increased opening pressure (>200 mm H_2_O), elevated protein levels (>50 mg/dL), decreased glucose concentrations (<60 mg/dL), and elevated white blood cell counts. (>5 cells/mm^3^). Although these abnormalities suggest CNS involvement, they are not definitive for diagnosis [23]. CSF cytology is estimated to have a specificity of >95% for detecting CNS involvement; however, it has a relatively low sensitivity of <50%, which can lead to false negative results. This low sensitivity is primarily due to the limited number of cells in the CSF and the morphological similarities that can complicate the distinction between benign and malignant cells [38]. FCM in the CSF has been utilized, either alone or in adjunctive cytology, to increase the sensitivity for detecting CNS involvement [39]. However, FCM for CSF analysis requires strict adherence to technical protocols. CSF must be collected via lumbar puncture, processed within one hour to prevent cell degradation, and concentrated using low-speed centrifugation. Some authors recommend the use of fixatives, such as TransFix/EDTA, to preserve cell integrity. FCM uses 6–9 monoclonal antibodies in the assay, further improving its sensitivity. However, the threshold for FCM positivity remains controversial. While some studies suggest a minimum of 30 events, others propose thresholds as low as 10 or 9 B cells [40,41,42,43].

Stable molecular biomarkers may help identify patients with CNSL and establish gold standards for CSF diagnostics. Recent studies have shown that CNS-derived circulating tumor DNA (ctDNA) and microRNAs are promising biomarkers for the ultrasensitive detection of CNSL from lymphomas and pediatric B-ALL [27,44,45].

Our pilot study is the first to evaluate the feasibility of using CD79A and IL7R mRNA expression in CSF as molecular diagnostic tools for detecting CNS involvement in adult patients with B-ALL. We found that 52.6% of the samples were positive for CD79A mRNA, 42.1% were positive for IL7R mRNA, and 26.3% were positive for both. Notably, IL7R expression levels were significantly higher in patients with MRI-detected CNS involvement (median fold change = 2.12) than in CNS-negative patients (median: 0.38), whereas CD79A expression showed a less pronounced trend (median: 0.62). This finding is consistent with previous studies demonstrating that IL7R protein is highly expressed in B-ALL cells collected from the CSF and bone marrow of pediatric patients with B-ALL [29]. Additionally, bone marrow samples from pediatric B-ALL patients who were CNS-positive presented significantly higher levels of CD79A protein than those from patients who were CNS-negative [30]. The observed 2.12-fold increase and statistical association between IL7R mRNA expression and patients with MRI-confirmed CNS involvement is biologically relevant, given the known function of IL7R in leukemic cell survival, proliferation, and tissue homing. IL7R signaling activates the JAK-STAT5, PI3K-AKT, and MAPK pathways, all of which are implicated in lymphoid malignancies [46,47]. Prior studies have shown that even relatively small increases in IL7R expression can potentiate these signaling cascades and alter the behavior of leukemic cells [29]. Given the inherently low cellularity of CSF, small increases in transcript abundance may reflect significant leukemic infiltration. Nevertheless, we recognize that a 2.12-fold change, while indicative, may not be sufficient for clinical diagnostic use.

In contrast, CD79A is a structural marker of B-lineage identity that is constitutively expressed in B-ALL blasts [48]. Its presence in the CSF may indicate the presence of leukemic cells, regardless of their functional state or invasive capacity. Moreover, its co-detection with IL7R may increase diagnostic confidence, especially in cases with borderline cytological and radiological findings.

In this study, the comparison of CD79A and IL7R mRNA expression levels in the CSF demonstrated that the median value for the CNS involvement group was slightly increased. In contrast, the range of values was more extensive in the non-CNS involvement group. This wide range could indicate significant variability within this group, which may originate from biological differences (e.g., blast cell infiltration dissimilarities) or other confounding factors. Consequently, this may reduce the statistical significance of the differences between the groups.

Several cytogenetic alterations have been associated with CNS involvement in patients with B-ALL, including t(1;19) TCF3-PBX1 and t(9;22) BCR-ABL1 fusion genes [49]. However, in the sample of adult patients analyzed in this work, only one patient was positive for t(9;22). In addition, the clinical characteristics of patients with B-ALL, such as leukocyte count, proliferative index, and lactate dehydrogenase levels, have also been proposed as risk factors for CNS relapses [23]. However, our bivariate analysis did not reveal any statistically significant associations between CNS infiltration and the clinical factors.

The small sample size may have contributed to this finding. The limited cohort (n = 19) reduced the statistical power, making it challenging to detect subtle associations. Small sample sizes inherently limit statistical power, which may cover subtle yet clinically meaningful associations between CD79A and IL7R mRNA expression and CNS involvement in a broader patient population. It also increases the probability of sampling bias due to variability in disease biology, clinical presentation, and patient characteristics [50]. Therefore, the data presented here should be viewed as exploratory and hypothesis-generating rather than offering definitive conclusions.

A significant limitation of this study is the absence of matched transcriptomic or FCM data from peripheral blood or BM blasts, which would have enabled a direct correlation between the systemic expression of CD79A and IL7R and their detectability in the CSF. This is particularly relevant given that only five out of 19 CSF samples were positive for both transcripts. Several factors may have contributed to this observation. First, CD79A and IL7R expression levels vary across B-ALL subtypes and disease stages [51], which may affect the probability of simultaneous detection. Second, leukemic infiltration into the CSF may be low or transient in some cases, resulting in transcript levels below the qPCR detection threshold. Third, the technical limitations inherent to CSF analysis, such as RNA instability or differential degradation, could influence the reliability of detecting specific transcripts. These findings underscore the need for future studies that integrate CSF and matched systemic data to clarify the diagnostic and biological significance of CD79A and IL7R expression in CNS involvement.

Another limitation of this study is the lack of consistently available long-term follow-up data (e.g., 24 months) for all patients in the cohort. Although patients were clinically monitored, the absence of standardized and comprehensive follow-up prevented a definitive assessment of whether lower-level transcript detection by qPCR could predict subsequent CNS relapses or progression. Without such outcome data, we cannot decisively confirm that the sensitivity of qPCR in our study exceeds that of conventional cytology. Consequently, our findings should be viewed as preliminary and intended to generate new hypotheses. Future studies with extended and standardized clinical follow-ups are crucial for determining the prognostic significance of qPCR positivity in CSF, especially in cases with negative cytology.

A limitation of our qPCR approach was the absence of a lineage-unrelated negative control gene. Although GAPDH was used as a reference for normalization, it did not account for cell-type specificity. This is particularly relevant for IL7R, which is also expressed on normal T lymphocytes that may be present in the CSF, independent of leukemic infiltration [52]. Consequently, some IL7R signals detected by qPCR may reflect physiological immune cell activity rather than true leukemic involvement. The inclusion of a negative control gene not expressed in lymphoid cells—such as a myeloid- or epithelial-specific transcript—would have helped clarify this distinction. Future studies should incorporate such controls to enhance the specificity and interpretative value of CSF qPCR-based diagnostics.

Moreover, although MRI findings are sensitive, they are not definitive for leukemic infiltration and should ideally be corroborated with advanced techniques such as high-sensitivity droplet-digital PCR (ddPCR) and flow cytometry.

More extensive multicenter studies are needed to validate the diagnostic utility of CD79A and IL7R mRNA levels in CSF for detecting CNS involvement in B-ALL and to investigate the potential of these biomarkers to predict CNS relapses or response to therapy. Integrating molecular markers with MRI and cytology may improve diagnostic accuracy by providing complementary information on leukemic infiltration, which could subsequently be incorporated into the diagnostic criteria system for leukemia infiltration into the CNS, according to the NCCN. MRI helps detect structural abnormalities, cytology identifies malignant cells, and molecular testing improves the sensitivity to detect minimal CNS involvement. “Combining these approaches will help develop more accurate diagnostic algorithms for CNS involvement in B-ALL.”

## 5. Conclusions

This pilot study presents promising preliminary data suggesting that CD79A and IL7R mRNAs could serve as potential biomarkers in the CSF for CNS involvement in adult B-ALL. Although the small sample size and exploratory nature of the study require careful interpretation of the results, this research establishes a foundation for future validation.

## Figures and Tables

**Figure 1 diseases-13-00206-f001:**
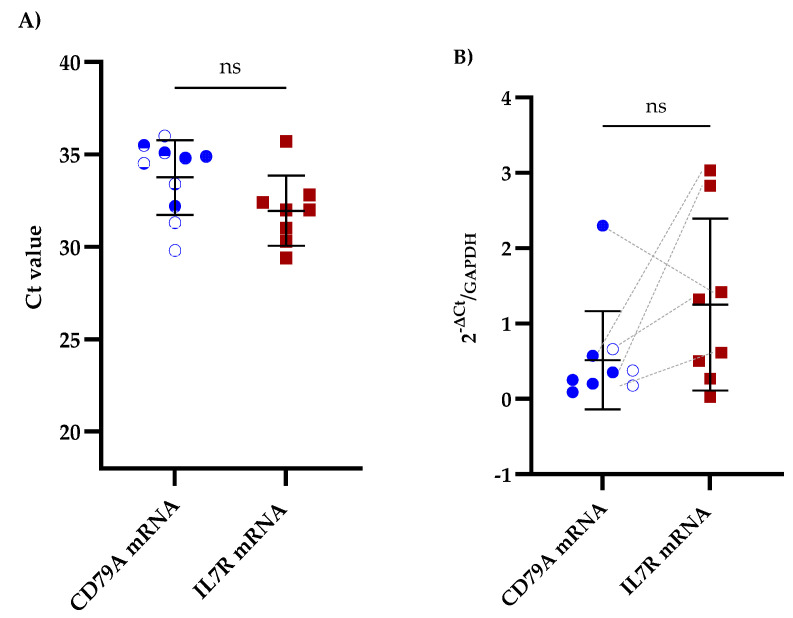
Comparative qPCR analysis of CD79A and IL7R levels in the CSF of patients with B-ALL. (**A**) Mean Ct values obtained via qPCR for CD79A and IL7R analyzed using the Kruskal‒Walli’s test and Dunn’s test for post hoc comparisons. (**B**) Relative expression of CD79A and IL7R mRNAs in the CSF analyzed using the Wilcoxon test; ns indicates no statistically significant difference. The five patients exhibiting concomitant expression of CD79A and IL7R are linked with grey dotted lines. GAPDH was used as a reference gene for the 2^−∆Ct^ analysis.

**Figure 2 diseases-13-00206-f002:**
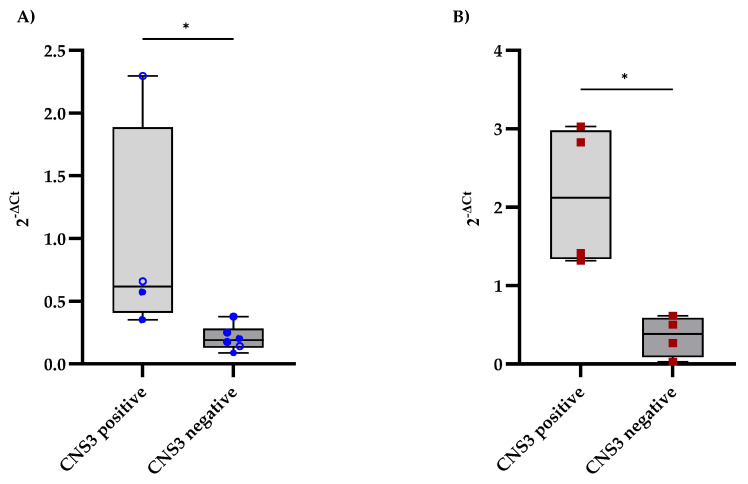
Comparison of CD79A and IL7R expression levels in CSF samples from B-ALL patients stratified according to the diagnostic criteria for CNS leukemia. (**A**,**B**) show the distributions of CD79A and IL7R expression, respectively, between patients with and without CNS involvement, as determined using the criteria in Table 1. Each qPCR reaction was performed in technical triplicate, and the intra-assay coefficient of variation for Ct values was consistently below 3%, supporting the reproducibility and stability of the measurement. The relative mRNA expression of CD79A and IL7R was calculated using the 2^−ΔCt^ method, with GAPDH as the reference gene. Statistical significance was assessed using the Mann‒Whitney test (* *p* < 0.05).

**Table 1 diseases-13-00206-t001:** Classification criteria for CNS Leukemia.

CNS Status	CSF Findings	Clinical Significance
CNS1 (Negative)	No detectable leukemic blasts in CSF (<5 WBC/μL and no blasts)	No CNS involvement
CNS2 (Indeterminate/Possible Involvement)	WBC < 5/μL but with detectable leukemic blasts in CSF	Uncertain significance may require close monitoring or treatment
CNS3 (Positive for CNS leukemia)	WBC ≥ 5/μL with leukemic blasts in CSF, or clinical/MRI signs of CNS disease	Confirmed CNS involvement, requires intrathecal therapy

Traumatic Lumbar Puncture (TLP): If the CSF contains ≥5 WBC/μL but blood contamination is suspected, correction formulas (e.g., adjusted WBC count) can help determine the true positivity. MRI findings (e.g., leptomeningeal enhancement) and clinical CNS signs may justify classification as CNS3, even with negative or inadequate cytology.

**Table 2 diseases-13-00206-t002:** Primer and probe sequences.

Gene	Forward Primer (5′→3′)	Reverse Primer (5′→3′)	Probe (5′→3′) FAM
CD79A	CCTGGGACATTCTCCTTTCA	CTGGCCTGGAGAAGAGTGAG	GCCCTTCTGGGGGCTTCCTT
IL7R	AGCCAGTTGGAAGTGAATGG	AGGCACTTTACCTCCACGAG	CGCAGCACTCACTGACCTGTGC
GAPDH	CAGCCTCAAGATCATCAGCA	TGTGGTCATGAGTCCTTCCA	CCCCTGGCCAAGGTCATCCA

FAM: Fluorescein amid.

**Table 3 diseases-13-00206-t003:** Clinical characteristics of naive B-ALL patients.

Code	F/M	Age (Years)	BMI (kg/m^2^)	WBC/mm^3^	Hb (g/dL)	PLT/mm^3^	PBB (%)	BMB (%)
1148684	F	30	38.5	261,400	12.7	38,000	58	96
1146429	F	18	34.4	1130	4.7	40,000	90	70
1170985	M	26	23	2450	5.2	74,000	23	90
1172165	M	22	31.14	98,590	6.7	1890	77	98
1182265	M	21	25.4	298,830	6.9	19,000	88	95
1182029	M	20	30.5	2080	5.7	35,000	60	92
1182016	F	27	19.1	14,830	8.6	18,000	60	98
1180010	F	24	24.4	1830	8	1460	10	70
1170637	M	18	22.4	1040	3.8	16,000	60	70
1163868	M	19	25.8	19,200	8.6	42,000	31	99
1150680	M	22	30.72	1090	5.8	123,000	0	91
1161154	F	31	32.4	83,320	4.2	30,000	95	95
1162218	M	43	33.4	2630	7	309,000	2	30
1162264	F	67	30.75	156,200	4.5	11,000	70	90
1158854	M	49	29.3	11,000	4.5	8000	87	90
1167915	F	46	24.9	77,530	5.8	10,000	86	86
1156718	M	55	31.25	1220	6.6	24,000	3	90
1167121	F	46	24.9	5090	6.8	28,000	58	92
1163863	F	56	24.6	14,600	3.6	18,000	91	98
mean ± sd (min-max)		33.6 ± 15.4 (17–67)	27.5 ± 5.1 (18–38.5)	55,477 ± 90,431 (1040–298,830)	6.3 ± 2.2 (3.6–12.7)	44,545 ± 70,014 (1460–309,000)	55.2 ± 33.5 (0–95)	86.3 ± 16.5 (30–99)

sd (min-max): standard deviation (minimum-maximum values), F/M: Female/Male, BMI: Body mass index, WBC: white blood cells, Hb: Hemoglobin, PLT: Platelets, PBB: peripheral blood blasts, BMB: Bone marrow blast.

**Table 4 diseases-13-00206-t004:** Bivariate analysis of factors associated with B-ALL infiltration into the CNS.

Variable	Category	Involvement in CNS (CNS3)		ORna ^a^	95%CI ^b^	*p*
		Positive	Negative			
Age ^c^	≤28.5	3	7	3.43	0.29–40.15	0.58
	>28.5	1	8			
Sex	Male	2	7	1.14	0.12–11.02	>0.99
	Female	2	8			
Leucocytes	Hyperleukocytosis ^d^	2	4	2.75	0.28–26.98	0.56
	Normal	2	11			
LDH ^e^	Increased	2	9	0.67	0.07–6.44	>0.99
	Normal	2	6			
Blast cells	Peripheral blood	3	12	0.75	0.05–10.70	>0.99
	Normal	1	3			
BMI ^f^	Overweight	2	9	0.67	0.07–6.44	>0.99
	Normal weight	2	6			
IL7R *	Positive	4	4	∞	—	0.02
	Negative	0	11			
CD79A *	Positive	4	6	∞	—	0.09
	Negative	0	9			

^a^ Unadjusted Odds Ratio; ^b^ 95% confidence interval; ^c^ years; ^d^ >100,000/µL; ^e^ Serum total lactic dehydrogenase, >350 UI/L; ^f^ BMI, body mass index. * Haldane–Anscombe correction and Fisher’s exact test.

## Data Availability

The data presented in this study are available upon request from the corresponding author. The data are not publicly available due to privacy or ethical restrictions.

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
