# Peer review of "CD79A and IL7R mRNA Levels in the Cerebrospinal Fluid of Adults with Acute B-Cell Lymphoblastic Leukemia: A Pilot Study"

_diseases, 2025, doi:10.3390/diseases13070206_

Round 1
Reviewer 1 Report
Comments and Suggestions for Authors
In the present manuscript, Salvatierra and collaborators explore the potential of IL-7R and/or CD79a expression levels in the cerebrospinal fluid (CSF) of B-ALL patients as biomarkers for central nervous system (CNS) involvement. However, the small number of patients enrolled in this study limits the ability to draw major conclusions and renders the studies preliminary. Moreover, conclusions should be improved since they do not accurately reflect the results presented.
Major criticisms
- The introduction lacks important references. The authors should cite the paper by Lenk et al., Blood. 2024. 143(26):2735-2748. doi: 10.1182/blood.2023021088. which has data demonstrating that IL-7R levels are also increased in patients bearing CRFL2 or TP53mut alterations.
- The authors should also cite the papers from Almeida et al., Nat Commun. 2021. 12, 7268. doi: 10.1038/s41467-021-27197-5, from Geron et al., Nat Commun. 2022 Feb 3;13(1):659. doi: 10.1038/s41467-022-28218-7, and from Thomas et al., Leukemia 2022. 36(1):42-57. doi: 10.1038/s41375-021-01326-x, when they mention that “By combining signals from IL7R and pre-BCR, B cells develop properly, but when these signals are disrupted in B-ALL, they lead to the sustained growth of leukemia cells”. In fact, this sentenced should be completed, since these three papers show that IL7R activation can not only sustain the growth of leukemia cells but actually drive leukemia development.
- Indicate in Table 1: a) which patients have CNS involvement (based on MRI and/or on cytology – please specify both); b) which patients are CD79a-positive by flow cytometry (supposedly all or the vast majority of them should be CD79a-positive); c) which ones are IL7R/CD127-positive by flow cytometry (at least four of them); d) cytogenetics of each patient (e.g. indicate the BCR::ABL-positive patient); and e) EGIL classification of each patient.
- The authors state that 5 of the patients display concomitant expression of CD79A and IL-7R. In Figure 1, please highlight them in a different colour in the graphs, so that it is possible for the reader to distinguish them from the others that do not have concomitant expression.
- In Figure 2, the authors do not specify whether a housekeeping gene was used for normalization in the qPCR analysis. This information is critical to ensure accurate and reliable quantification of CD79A and IL7R expression levels. In addition, the authors could try to do the same approach but to correlate for each patient CD79A and IL7R levels.
- Table 4 should include the statistical values relative to CD79A and IL7R.
- Unclear what is the relevance of comparing CD79A and IL7R levels in Figure 1.
- In the abstract and parts of the main text, the authors place the emphasis on IL7R being a potential biomarker for CNS involvement, neglecting CD79A. This is hard to understand, given that both IL7R and CD79A mRNA expression in the CSF show a statistical association with CNS involvement in univariate analysis (Figure 2) and none of them apparently shows statistical significance (although actual values are not presented in Table 4). Conclusions should be corrected to clarify that both IL7R and CD79A showed an association with CNS involvement in univariate analysis but neither IL7R nor CD79A served as independent biomarkers of CNS involvement in bivariate analysis.
- CD79a should be expressed in most B-ALL cases. Therefore, detection of CD79A mRNA in the CSF of patients with CNS involvement does not implicate that CD79a contributes to CNS involvement but, more likely, it merely reflects the fact that there are B-ALL cells in the CNS and therefore CD79A transcripts. This does not affect the potential use of CD79A transcript levels as biomarkers for CNS involvement but statements such as “More extensive multicenter studies are needed to confirm the role of CD79A and IL7R mRNAs in CNS infiltration” should be avoided.
- Importantly, the authors should present data comparing whether CD127 and CD79a surface levels on patients’ blasts collected from the bone marrow or blood by flow cytometry are associated with CNS involvement. This would be a much more accurate manner of understanding potential contributions of each gene for CNS spread, independently of their value as biomarkers.
Minor criticisms
- The article would benefit from proofreading.
Comments on the Quality of English Language
Overall, the quality of the written English is acceptable, although there are errors here and there. Proofreading by a native speaker is advised.
Author Response
"Please see the attachment."

Reviewer 2 Report
Comments and Suggestions for Authors
Reliable detection of the presence of leukemic blasts in CSF from patients with ALL remains challenging and profoundly limits the prognostic strength of the findings. In this study, Salvatierra and colleagues report the detection of CD79A and IL7R transcripts in CSF obtained prior to initiation of treatment from 19 adults with ALL. Through comparisons with CNS-ALL detection using other methods, the authors are able to show that transcript levels correlate with CNS involvement and propose that they could be used as biomarkers to inform therapy.
Strengths
This is an important question to address
The authors have built a reasonably sized cohort, given the relative rarity of adult ALL.
The authors acknowledge and discuss the limitations of their study
The manuscript is generally well written, and progresses logically
The comparison of methods used to assess CNS involvement greatly adds to the study
While somewhat limited, the results presented strongly suggest that this approach should be further investigated.
Weaknesses
The authors are not the first to use PCR for detection for CSF-ALL – previous studies should be cited in the Introduction.
It would greatly help interpretation of the CSF PCR results if expression (transcript or protein) of CD79a and IL7R on the patient-matched peripheral or BM blasts was provided. In the absence of that data, the authors should provide some discussion as to why only 5/19 CSF samples were positive for both transcripts.
A second table to present the study findings would add significantly to the manuscript. This table should include ALL classification, clinical CNS status, cytomorphological CNS status, MRI-defined CNS status, PCR-defined status, and latest follow-up status for each patient. This would allow the reader to identify the patterns and consistency of the various detection methods.
PCR on CSF samples from non-leukemia patients would help interpretation of the patterns detected in the ALL patients…but I acknowledge that such sample may not be available.
Inclusion of a negative control gene for qPCR would strengthen the results, especially considering IL-7R is also expressed on T cells, which are also likely present in the CSF.
There should be up to 2 years of follow-up for the patients on this study. Inclusion of that data could help with the interpretation of the clinical relevance of lower transcript level detection. In the absence of such information, the sensitivity of qPCR does not appear to be any higher than cytology and this pojnt should be discussed.
Minor points
While the manuscript is generally well written, the Introduction is somewhat disjointed, repetitive, and contains information of peripheral relevance to the study. I would suggest editing it to better focus on the pertinent clinical and biological background.
Line 60: “Leukemic cells can move into CNS by entering the CSF.” Without citation, this does not add anything. Blasts can enter via several potential (but mostly unproven) pathways.
Line 80: Abbreviation for flow cytometry used here (FCyt) is different form the one used later (FC).
Line 154 and 174: Combine the two sections on MRI
Line 274: “Finally, according to classification of CNS leukemia (Table 3), four patients were classified as CNS3”. Should refer to Table 2.
Line 410: “More extensive multicenter studies are needed to confirm the role of CD79A and IL7R mRNAs in CNS infiltration”. This study does not provide any evidence of the role of CD79A and IL7R in CNS infiltration – it uses them only as a target for detection.
Line 417: Why is the last sentence in inverted commas?
Author Response
"Please see the attachment."

Reviewer 3 Report
Comments and Suggestions for Authors
This study analyzed mRNA expression in the cerebrospinal fluid (CSF) of patients with B-cell acute lymphoblastic leukemia (B-ALL) to evaluate the potential of IL17R and CD79A as markers of central nervous system (CNS) involvement. The research presents a clear objective and a well-structured experimental plan with attention to detail. Both the results and discussion are well-articulated. Only minor suggestions for improvement are provided as follows.
The abstract is well written, and the introduction effectively presents the research background and clearly states the objective. One suggestion for improvement is to expand the second paragraph by elaborating on how early detection of CNS involvement can benefit treatment outcomes. Providing more detail on this point would strengthen the rationale for the study and highlight its clinical relevance.
The methodology is well described. RNA degradation, which would typically be a major concern in this type of analysis, has been appropriately addressed through careful experimental design and quality control measures to assess RNA integrity.
There is a minor typographical error on line 127—the reference citation appears outside the period at the end of the sentence.
The results and discussion are well structured and clearly presented. To enhance the utility of this work for other researchers interested in mRNA markers in CSF, it would be helpful to include data on the quality of the retrieved mRNA, as well as the cycle threshold (Ct) values for housekeeping genes such as GAPDH.
Author Response
"Please see the attachment."

Reviewer 4 Report
Comments and Suggestions for Authors
Minor Comments-
- Abstract contains grammatical errors and awkward phrasing (e.g., “The purpose of the study was investigated…”). Please revise the abstract for clarity and grammatical correctness.
- Current study presents a potentially valuable biomarker (IL7R mRNA) for CNS involvement in adult B-ALL, but the sample size (n=19) is relatively small. The authors should acknowledge this as a limitation and discuss how it affects the statistical power and generalizability of the findings.
- The retrospective design is appropriate, but additional details on inclusion/exclusion criteria and how data were extracted should be provided for reproducibility.
- Diagnostic criteria for CNS involvement via MRI and clinical assessments should be clearly defined. How was CNS disease confirmed in the absence of cytological evidence in some patients?
- The study does not comment on CD79A mRNA expression findings. Was there no significant difference, and if so, what does that imply about its potential utility?
- Difference in IL7R mRNA expression is statistically significant, but the biological relevance of a 2.12-fold change should be discussed. Is this threshold sufficient for diagnostic application?
- The acronym "naive B-ALL patients" in the results section should be clarified. Does it refer to newly diagnosed, treatment-naïve patients?
- Manuscript is generally well-written, though there are occasional grammatical issues and passive constructions that could be revised for clarity.
- The discussion should explore why IL7R, but not CD79A, showed differential expression and what this implies for the pathophysiology of CNS infiltration in B-ALL.
- Figures are clear, though the legends should be more detailed for standalone interpretation.
- Some references are outdated; include more recent studies, especially post-2023, to contextualize findings within the latest research.
Author Response
"Please see the attachment."

Round 2
Reviewer 1 Report
Comments and Suggestions for Authors
The authors have properly addressed previous criticisms.